# Ap4A in Cancer: A Multifaceted Regulator and Emerging Therapeutic Target

**DOI:** 10.3390/molecules30153056

**Published:** 2025-07-22

**Authors:** Kateryna Tkachenko, Tiziana Bachetti, Camillo Rosano

**Affiliations:** Proteomics and Mass Spectrometry Unit, IRCSS Ospedale Policlinico San Martino, 16132 Genova, Italy; kateryna.tkachenko@hsanmartino.it (K.T.); tiziana.bachetti@hsanmartino.it (T.B.)

**Keywords:** diadenosine tetraphosphate, cancer, cellular process

## Abstract

Diadenosine polyphosphates, including diadenosine tetraphosphate (Ap4A), are ubiquitous nucleotides that are present across diverse life forms, gaining considerable interest due to their role as cellular signaling molecules. Ap4A, in particular, has been extensively researched in various biological systems, especially under conditions of environmental stress. This review provides an in-depth analysis of the current knowledge surrounding Ap4A, focusing on its biosynthesis and degradation pathways, the identification of Ap4A protein targets and the molecular mechanisms underlying its action. Furthermore, this review aims to examine the interplay between the various pathogenetic mechanisms driving tumor development and the potential role of Ap4A which emerges as pivotal signaling molecules orchestrating cellular responses to environmental challenges, positioning them at the nexus of cancer adaptation and progression.

## 1. Introduction

In the intricate landscape of tumor biology, cancer cells are continually exposed to a variety of intrinsic and extrinsic stressors, including hypoxia, nutrient deprivation, oxidative damage, and immune surveillance. To survive and thrive under these hostile conditions, tumor cells develop sophisticated adaptive strategies that involve dynamic reprogramming of both metabolic and transcriptional processes. One such adaptation involves the use of molecular “alarmones”—small signaling nucleotides that enable rapid cellular responses to stress. Among these, diadenosine tetraphosphate (Ap4A) has emerged as a compelling candidate. First discovered in 1966, Ap4A was long considered a metabolic by-product; however, it is now recognized as a potent stress signal involved in diverse physiological and pathological processes. Much like bacterial alarmones such as guanosine tetraphosphate (ppGpp), which orchestrate wide-ranging adjustments to cellular activity during nutrient stress, Ap4A has been shown to regulate proliferation, survival, and gene expression in mammalian cells [1].

Ap4A functions as both a signaling molecule and a regulator of cellular homeostasis, particularly under stress [2]. It is tightly regulated within cells and can accumulate in response to stressors, triggering protective mechanisms to preserve cellular integrity [3]. Early studies demonstrated that Ap4A levels vary significantly across mammalian cell lines, with higher levels observed in rapidly proliferating cells and an inverse correlation with doubling time—indicating its potential role as a metabolic signal linked to cell growth and division [4].

One of Ap4A’s central roles lies in the regulation of the cell cycle. By interacting with key proteins, it influences transitions through various cell cycle phases, ensuring controlled proliferation. Disruptions in this regulatory function could contribute to pathological conditions characterized by unregulated cell division, including cancer [5].

In the context of tumor biology, Ap4A’s role becomes even more intriguing. Tumorigenesis is driven by a multitude of factors—genetic mutations, aberrant signaling pathways, metabolic reprogramming, and interactions with the tumor microenvironment (TME) [6]. Tumor cells must navigate and adapt to these pressures, and Ap4A appears to play a critical role in this process. Though its mechanisms in cancer are still being unravelled, emerging evidence points to Ap4A as a key player in tumor cell adaptation and survival.

Beyond cancer, its role has been widely studied in other systems. For instance, it is co-stored with ATP in the dense granules of platelets and released upon activation in the vascular system. Alongside Ap2A and Ap3A, Ap4A contributes to cardiovascular regulation by inducing vasodilation through the stimulation of nitric oxide (NO) and prostacyclin (PGI_2_) production in endothelial cells [7]. These findings highlight Ap4A’s broader signaling functions in inflammation, vascular tone, and endothelial homeostasis. Notably, Ap4A interacts with purinergic receptors, including several P2X and P2Y subtypes. Although often a partial agonist compared to ATP, Ap4A modulates purinergic signaling pathways, including calcium flux, and may temper excessive purinergic activation [8]. This modulation becomes particularly relevant in cancer, where Ap4A levels rise in response to oxidative and thermal stress—suggesting a role in tumor progression, immune evasion, and resistance to therapy.

This review delves into the multifaceted roles of Ap4A in both normal and cancerous contexts, with a particular emphasis on its involvement in key hallmarks of cancer such as apoptosis evasion, metabolic reprogramming, stress adaptation, and proliferative signaling. We explore its interactions with purinergic receptors—most notably P2X7—and examine how these relationships influence tumor biology and may serve as targets for therapeutic intervention. Finally, we address current challenges and future perspectives in leveraging Ap4A’s biology for cancer treatment.

## 2. AP4A Structural Conformation

All kingdoms of life have a family of nucleotides known as Dinucleoside Polyphosphates (NPnN; N = adenosine, guanosine, uridine, or cytidine, and n = number of phosphates) [9]. Since their discovery, two nucleosides joined by an arbitrary number of phosphate groups have been found in a variety of organisms [10]. Among possible NpNs, diadenosine tetraphosphate (Ap4A), which consists of two adenosine moieties connected by phosphoester linkages at their 5′-hydroxyl groups to form a chain of four phosphates, is one of the most prominent examples (Figure 1) [11].

The significance of Ap4A was first recognized in the 1960s, during studies of brine shrimp eggs, where its role in DNA synthesis was observed [12]. Subsequent research revealed its involvement in biochemical processes related to tRNAs and amino acids during protein synthesis. Extensive investigations into Ap4A’s structural and chemical properties have demonstrated that different factors such as temperature and pH substantially affect its conformation [13,14]. Thus, Ap4A can adopt three primary conformations: stacked, folded unstacked, and open [15]. In the stacked conformation, the adenine rings are aligned and stacked while the phosphate chain is folded. In the folded unstacked conformation, the phosphate groups are folded, but the adenine rings are oriented in opposite directions and are not stacked. The open conformation features a fully extended phosphate chain and unstacked adenine rings. For a detailed depiction of the conformational arrangement discussed below, please refer to Stern et al. [15], where the structure is extensively illustrated and described. These conformational changes are influenced by factors such as pH, temperature, and metal ions. At physiological pH, Ap4A predominantly assumes a stacked conformation due to electrostatic interactions between the partially positively charged adenine rings and the negatively charged phosphate groups. In slightly acidic conditions (pH 4–5), Ap4A may adopt an unstacked conformation, while at very low pH (below 3), protonation of the phosphate groups reduces electrostatic attraction, resulting in an open conformation. In addition, the unstacked conformation also occurs with increases in temperature. Thus, these conformations enable Ap4A to interact with various purinergic receptors, which may be present in environments with different pH and ion concentrations [16].

## 3. Regulation of Intracellular Ap4A Levels

The cellular balance of Ap4A is tightly regulated through its biosynthesis and degradation, ensuring its optimal function as a stress-responsive signaling molecule. While aminoacyl-tRNA synthetase drives its synthesis under normal conditions, degradation by specific hydrolases such as Nucleoside Diphosphate Linked Moiety X-Type Motif 2 (Nudix hydrolase 2) prevents its pathological accumulation, highlighting a delicate equilibrium.

### 3.1. Biosynthesis of Ap4A

Ap4A is synthesized primarily through the action of the aminoacyl-tRNA synthetase (aaRS) during the process of tRNA charging. This enzyme catalyzes the formation of Ap4A by transferring the adenylate moiety from ATP to an already adenylated nucleoside diphosphate, such as ADP, resulting in the production of Ap4A [11]. This reaction typically occurs as a by-product of the tRNA aminoacylation process, where amino acids are attached to their corresponding tRNA molecules. In proliferating cells and cancer tissues, aaRS expression and activity are frequently upregulated, correlating it with increased Ap4A synthesis. In immune cells, lysyl-tRNA synthetase (LysRS) can be phosphorylated upon immune activation (e.g., in mast cells or macrophages), causing a shift from translation-related functions to Ap4A synthesis [5]. This process is critical for regulating immune gene expression, such as MITF and other transcription factors [17]. In neurons and platelets, Ap4A is synthesized in response to stimulation and stored in synaptic vesicles or dense granules, respectively. These specialized cells rely on stimulus-dependent exocytosis to release Ap4A into the extracellular environment, where it can act as a purinergic signaling molecule.

In addition to aaRS, other enzymes are also capable of synthesizing Ap4A, including DNA ligases and certain enzymes involved in DNA repair pathways [18]. For instance, DNA ligases can catalyze the synthesis of Ap4A from ATP and ADP during DNA replication and repair processes [10]. The production of _Ap4A_ in these contexts is often associated with cellular stress responses, where the molecule functions as a signaling mediator involved in regulating cell growth, apoptosis, and oxidative stress responses [19]. Ap4A synthesis is tightly regulated and varies depending on the cell’s metabolic state and external stress conditions. Under normal physiological conditions, intracellular Ap4A levels remain low. However, they can rise significantly in response to environmental stressors such as heat shock, oxidative damage, or genotoxic stress [19]. In Drosophila cell cultures, for instance, heat shock and heavy metal exposure led to a 2–3-fold increase in Ap4A levels compared to unstressed controls [20]. Similarly, in the slime mold Physarum, hypoxia induced a 3–7-fold rise in Ap4A, which rapidly returned to baseline upon reoxygenation—illustrating its dynamic sensitivity to metabolic stress [21]. Conversely, during growth-inhibitory conditions such as serum deprivation or amino acid starvation, Ap4A levels can decline dramatically—by up to 100-fold—despite stable intracellular ATP concentrations. This divergence suggests that Ap4A operates as a distinct metabolic “alarmone,” finely tuned to signal adverse conditions and initiate adaptive responses. Therefore, the predominant mode of Ap4A synthesis—whether constitutive or inducible, enzymatic or stress-regulated—varies according to cell type and functional context.

### 3.2. Degradation of Ap4A

Ap4A levels are also tightly regulated by specific degradation pathways. The degradation of Ap4A is essential for maintaining cellular nucleotide balance and preventing its accumulation. Thus, Ap4A hydrolase (ApaH) is a principal enzyme responsible for Ap4A degradation, and it is also known as bis(5’-nucleosyl)-tetraphosphatase (asymmetrical), encoded in humans by the *NUDT2* gene [9]. It was observed a 175-fold increase in the intracellular Ap4A by deleting *NUDT2* [3] confirming its significance as a major Ap4A hydrolase. This enzyme hydrolyzes Ap4A into AMP and ATP or into two molecules of ADP, depending on the organism and the specific hydrolase involved [9], which could otherwise lead to dysregulated signaling pathways and cellular dysfunction. Ap4A can also be targeted by other nucleotidases, such as Nudix family enzymes, which have broad substrate specificity and can hydrolyze various nucleotide diphosphates. By generating mutant strains defective in Ap4A-degrading enzyme(s), the accumulation of Ap4A was observed at high intracellular levels in different bacteria, highlighting Ap4A pleiotropic effects, such as loss of motility, higher heat and oxidative stress sensitivity or lower antibiotic tolerance [22]. Recently, a recombinant protein YqeK (COG1713) from Staphylococcus aureus copurified with ADP was identified as a class of symmetrically cleaving Ap4A hydrolases, showing a catalytic efficiency toward Ap4A similar to that of the symmetrically cleaving Ap4A hydrolases of the known ApaH family [23]. Overall, the degradation of Ap4A by specific hydrolases and nucleosidases is a critical aspect of cellular homeostasis, ensuring that Ap4A functions appropriately within its signaling and regulatory roles. In cancer cells, degradation is often reduced due to the downregulation or mutation of hydrolases, leading to the accumulation of Ap4A, which can interfere with tumor suppressors (e.g., HINT1) and modulate gene expression.

## 4. Role of Ap4A: Functions in Normal and Cancer Cells

Diadenosine tetraphosphate (Ap4A) is a multifunctional signaling molecule that plays crucial roles in maintaining cellular homeostasis and responding to environmental challenges. While traditionally considered a metabolic by-product, Ap4A is now recognized as an active regulator of diverse biological processes, including cell proliferation, stress response, and apoptosis. Its intracellular concentration is tightly regulated under physiological conditions, but can fluctuate dramatically in response to stress or pathological stimuli, such as DNA damage, nutrient deprivation, or oncogenic transformation. In cancer, the dysregulation of Ap4A levels—whether due to altered synthesis, hydrolysis, or receptor interactions—has been implicated in several hallmark processes of tumorigenesis. This section explores the functional roles of Ap4A in both normal and cancerous contexts, focusing on its involvement in apoptosis evasion, metabolic reprogramming, stress response modulation, control of cell growth and proliferation, and signaling through purinergic receptors, with particular attention to the P2X7 receptor’s role in cancer progression.

### 4.1. Apoptosis Evasion

Ap4A plays a crucial role in regulating apoptosis, the process of programmed cell death vital for maintaining cellular balance and development [24]. Apoptosis eliminates damaged, dysfunctional, or unnecessary cells through tightly controlled mechanisms, and Ap4A is a significant regulator in this context.

Under certain conditions, Ap4A promotes apoptosis, often through activating caspases [25], proteases critical for the execution of cell death. Ap4A enhances caspase activity, leading to the dismantling of cellular components and cell death, which is essential for removing cells with irreparable DNA damage and preventing malignancies. Moreover, Ap4A influences apoptosis by modulating various apoptotic pathways through its interaction with purinergic receptors, particularly P2X and P2Y families. By binding to these receptors, Ap4A triggers intracellular signals that can either promote or inhibit apoptosis, depending on the cellular context and receptor subtype. Conversely, Ap4A can also exhibit anti-apoptotic effects in other scenarios. For instance, under moderate stress, Ap4A may activate survival pathways like the PI3K/AKT signaling pathway [26], promoting cell survival and protecting against premature apoptosis. This ability to balance pro-apoptotic and anti-apoptotic effects allows Ap4A to fine-tune cellular responses to environmental conditions and internal signals, maintaining cellular integrity and function (Figure 2). Tumor cells may exploit Ap4A’s anti-apoptotic properties to evade destruction, highlighting the potential for targeting Ap4A-mediated pathways in cancer therapy.

### 4.2. Metabolic Adaption

One of the primary ways Ap4A contributes to metabolic adaptation is by regulating cellular energy homeostasis. Ap4A interacts with various enzymes involved in ATP synthesis and degradation, ensuring that cells maintain an adequate energy supply while preventing excessive energy expenditure. This regulation is particularly important in cancer, where energy demands are elevated due to increased cell division and biosynthesis. Ap4A’s involvement in managing ATP levels helps cancer cells adapt to the high-energy requirements of rapid proliferation and growth [27].

Ap4A accumulates under stress conditions such as oxidative stress, heat shock, and DNA damage. So, indirectly, increased Ap4A may promote AMPK activation if it contributes to a stress response, elevates intracellular calcium, or causes mitochondrial dysfunction leading to ATP depletion.

Moreover, Ap4A affects several metabolic pathways by modulating key enzymes and regulatory proteins. For instance, Ap4A can influence the activity of AMP-activated protein kinase (AMPK), a central regulator of cellular energy balance. AMPK activation, in turn, affects pathways involved in glucose uptake, fatty acid oxidation, and protein synthesis, all of which are critical for supporting cancer cell metabolism and growth [28]. Ap4A’s role in modulating AMPK activity underscores its importance in maintaining energy balance and supporting the anabolic processes required for tumor expansion.

In addition to its role in regulating apoptosis, the PI3K/AKT pathway is also critically involved in metabolic adaptation. Notably, Ap4A has been shown to influence the PI3K/AKT signaling cascade, which governs key cellular processes such as glucose metabolism, lipid synthesis, and cell growth. Although the precise molecular mechanism by which Ap4A affects PI3K/AKT signaling remains to be fully elucidated, emerging evidence suggests that Ap4A may indirectly influence this cascade by modulating upstream purinergic receptors or cellular stress responses. For instance, Ap4A interaction with purinergic receptors such as P2X7 can lead to downstream activation of PI3K, promoting AKT phosphorylation and supporting oncogenic signaling. Given that dysregulation of this pathway is a hallmark of many cancers, Ap4A’s ability to modulate PI3K/AKT signaling may play a significant role in driving the metabolic reprogramming characteristic of tumor cells [29]. High Ap4A levels can promote metabolic adaptations that favour tumorigenesis, including increased glucose uptake, enhanced mitochondrial function, and altered lipid metabolism. These adaptations enable cancer cells to thrive in challenging conditions and continue proliferating despite metabolic stress.

In the context of nutrient deprivation, Ap4A facilitates metabolic adaptation by modulating cellular responses to low glucose or other essential nutrients. During such conditions, Ap4A can influence signaling pathways related to autophagy, a process through which cells degrade and recycle their own components to maintain energy levels and remove damaged organelles. By promoting autophagy, Ap4A helps cancer cells adapt to nutrient scarcity, ensuring their survival and continued proliferation even in nutrient-limited environments [27]. This adaptation is crucial for tumor growth, especially in the nutrient-poor microenvironments often found within solid tumors. The ability of Ap4A to regulate these processes highlights its potential as a therapeutic target for disrupting the metabolic advantages of cancer cells.

### 4.3. Stress Response Modulation

Cancer cells frequently encounter diverse forms of stress, including oxidative stress and changes in the extracellular environment. Their ability to effectively respond to these stressors is crucial for their survival, proliferation, and resistance to therapy.

In response to stress, Ap4A levels rise rapidly, triggering defensive mechanisms that enhance cell survival. Elevated Ap4A levels are believed to mitigate oxidative damage caused by reactive oxygen species (ROS) from UV light or toxins [2]. Additionally, Ap4A levels increase during heat shock, aiding cells in managing thermal stress by facilitating protein repair and stabilization. Therefore, Ap4A induces the expression of Heat Shock Proteins (HSPs), which assist in refolding denatured proteins and degrading irreversibly damaged ones [10,19]. In addition, stress response involves the upregulation of antioxidant defences and repair mechanisms, including the activation of nuclear factor erythroid 2-related factor 2 (Nrf2), which orchestrates the expression of genes involved in detoxifying ROS and repairing oxidative damage [30].

Furthermore, Ap4A’s role in stress response extends to its interaction with purinergic receptors, such as P2X and P2Y receptors. These receptors are involved in various intracellular signaling pathways that regulate cellular stress responses. For example, Ap4A, by binding to P2X receptors, can activate the ATP-sensitive K+ channels, influencing cellular ion homeostasis and helping cells manage stress-induced changes in membrane potential and intracellular calcium levels. Similarly, Ap4A’s interaction with P2Y receptors can modulate intracellular signaling cascades, such as the MAPK/ERK and PI3K/AKT pathways, which are involved in stress response and survival signaling [31].

Cancer cells’ ability to withstand and adapt to stress is a hallmark of their malignancy and contributes significantly to their resistance to conventional therapies. Elevated levels of Ap4A in cancer cells often correlate with increased resistance to chemotherapy and radiation therapy, as the enhanced stress response mechanisms facilitate cancer cell survival and repair in the face of therapeutic insults. For instance, Ap4A can promote DNA repair pathways that counteract the damage induced by chemotherapeutic agents, thereby reducing their efficacy and contributing to the development of therapy-resistant tumors [2].

In summary, Ap4A plays a crucial role in stress response modulation in cancer cells by enhancing their ability to survive and adapt to various forms of stress. Understanding Ap4A’s role in these processes may provide new insights into developing strategies to overcome cancer cell resistance and improve treatment outcomes (Figure 2).

### 4.4. Cell Growth and Proliferation

Cancer cells are known for their ability to undergo significant metabolic reprogramming to sustain their rapid growth and proliferation. This reprogramming is crucial for meeting the high demands for energy and biosynthetic precursors required for tumor development. Thus, Ap4A plays a pivotal role in this metabolic adaptation, influencing both energy homeostasis and the activity of metabolic enzymes. In cancer cells, Ap4A levels are often dysregulated, leading to excessive cell division and tumor growth, suggesting its role in malignant transformation and proliferation. In addition, emerging evidence suggests that Ap4A contributes to tumor development by promoting angiogenesis, facilitating metastasis, and enhancing resistance to apoptosis [24].

For instance, Ap4A modulates cyclin-dependent kinases (CDKs) and other regulators, affecting cell proliferation. Cell cycle progression is a tightly regulated process orchestrated by a sequential activation of CDKs and their regulatory partners, the cyclins. The cell cycle consists of four main phases: G_1_ (cell growth), S (DNA synthesis), G_2_ (preparation for mitosis), and M (mitosis). Progression through these phases depends on the activation of specific cyclin–CDK complexes, which are themselves regulated by phosphorylation, proteolysis, and CDK inhibitors [32].

In normal cells, two major checkpoints govern the fidelity of cell division: the G_1_/S checkpoint, ensuring DNA integrity before replication, and the G_2_/M checkpoint, which prevents mitotic entry in the presence of damaged or unreplicated DNA [33]. A key feature of tumor cells is the decoupling of DNA damage from checkpoint responses. While normal cells undergo senescence or apoptosis when checkpoint fidelity fails, tumor cells with defective checkpoints continue to divide, accumulating mutations that drive oncogenesis [34,35]. Thus, in breast carcinoma cell lines, it was found that knocking down *NUDT2* led to reduced cell proliferation and an accumulation of cells in the G_0_/G_1_ phase of the cell cycle, further highlighting Ap4A’s role in regulating cancer cell growth [36].

Further research showed that by inhibiting WNT/β-catenin pathway, which regulates cell proliferation, differentiation, and migration, Ap4A can suppress the transcription of genes that promote tumor growth and metastasis [37]. Through its ability to downregulate WNT/β-catenin signaling, diadenosine tetraphosphate exerts an additional level of control over oncogenic pathways, further positioning it as a key regulator of cancer progression.

Early studies proposed that Ap4A, synthesized by DNA ligase III, delays the initiation of new DNA replicons (the starting points for DNA replication). This delay is believed to prevent replication, giving cells time to repair potential DNA damage, particularly in ovarian cancer cell lines [3]. Such a mechanism could limit uncontrolled proliferation, positioning Ap4A as a potential negative regulator of cell cycle progression in ovarian cancer. Ap4A is synthesized under stress conditions, such as DNA damage, where it inhibits the initiation of DNA replication, contributing to the maintenance of genomic integrity [3,38]. DNA ligase III synthesizes Ap4A, which acts as a primer for ADP-ribosylation by PARP. The ADP-ribosylated Ap4A binds to the Ap4A-binding site on the protein associated with DNA polymerase-a-primase and inhibits it, possibly by reducing its interaction with DNA. ADP-ribosylated Ap4A also diffuses to other replication forks and inhibits DNA polymerase-a-primase there [39]. It was observed that cells depleted of certain DNA repair proteins such as poly (ADP-ribose) polymerase (PARP1), showed increases in Ap4N (N = adenosine, guanosine, uridine, or cytidine), up to 14-fold Ap4A, preventing replication fork progression.

### 4.5. Interaction with Purinergic Receptors

As previously mentioned, the interaction between Ap4A and purinergic receptors has several important implications in the oncological context. Diadenosine polyphosphates, when present extracellularly, can bind to and activate various P2X and P2Y receptor subtypes responsive to ATP across multiple mammalian tissues, eliciting a broad range of physiological effects [40]. The alarmone and purinergic receptors are known for their role in regulating various physiological processes, including inflammation, immune response, and cellular communication. In particular, the interaction between Ap4A and these receptors influence the TME through mechanisms that modulate immune cell infiltration, angiogenesis, and metastatic potential. However, the purpose of this review is not to explore into the functions of purinergic receptors—an extensively studied topic covered in excellent reviews and specific studies—but rather to provide an overview of Ap4A’s involvement in these dynamics.

#### 4.5.1. P2X Receptors

P2X receptors are a family of ligand-gated ion channels that are activated by extracellular ATP. These receptors are integral membrane proteins composed of three subunits, which can form either homo- or heterotrimers. Upon binding ATP, these channels open, allowing the flow of cations such as sodium (Na^+^), potassium (K^+^), and calcium (Ca^2+^), leading to depolarization of the cell membrane and subsequent intracellular signaling events. P2X receptors are known to be involved in various physiological and pathological processes, including synaptic transmission, inflammation, pain perception, and immune responses [31]. For instance, P2XRs are expressed by multiple malignant tumors and are increasingly recognized as prognostic indicators and potential therapeutic targets [41]. Additionally, P2X4, P2X5, and P2X7 receptors exist on the membrane of multiple tumor cells, such as non-small cell lung cancer, colorectal cancer, bladder cancer, renal cancer, as well as hematological malignancies, to promote the proliferation and metastatic potential of the tumor [42]. Ap4A acts as an agonist at several P2X receptors; therefore, its specific interactions with key P2X receptor subtypes and their roles in cancer will be discussed below.

#### 4.5.2. P2X1 Receptors

Ap4A acts as an agonist at P2X1 receptors, which are predominantly expressed on endothelial cells. The activation of P2X1 by Ap4A can promote endothelial cell proliferation and migration, key processes in angiogenesis, which is crucial for tumor growth as it provides the blood supply necessary for rapid cancer cells proliferation [43]. Thus, by facilitating angiogenesis, the activation of P2X1 receptors supports tumor expansion, making it a potential therapeutic target in inhibiting the vascularization of tumors.

#### 4.5.3. P2X4 Receptors

Similarly, to P2X1, P2X4 receptors also plays a significant role in endothelial cell function and angiogenesis, when activated by Ap4A, contributing to the proliferation and migration of endothelial cells, processes essential for the formation of new blood vessels within the tumor microenvironment. This activation supports the development of a vascular network that sustains tumor growth and metastasis. Additionally, P2X4 receptors are involved in the release of pro-angiogenic cytokines, which further enhances the vascularization of tumors, promoting their survival and expansion in a hypoxic environment [27].

#### 4.5.4. P2X7 Receptors

Recent evidence suggests that diadenosine tetraphosphate contributes to tumor development and immune modulation through its interaction with purinergic receptors, particularly P2X7. While Ap4A is not a primary agonist for P2X7 receptors, it can modulate their activity indirectly, which plays a significant role in the TME. Activation of P2X7 in tumor cells has been shown to trigger a range of downstream signaling pathways, including PI3K/AKT, JNK, and Wnt/β-catenin cascades, which collectively support tumor cell survival, proliferation, invasion, and angiogenesis. Specifically, this receptor modulates inflammatory cytokine production, such as IL-1β and TNF-α, often through calcium-dependent signaling, and can promote the expression of matrix metalloproteinases like MMP2, facilitating tumor metastasis [44]. Furthermore, P2X7 activation can lead to the formation of membrane pores, promoting cell death in immune cells and contributing to immune evasion by tumors. Thus, Ap4A’s indirect modulation of P2X7 receptors may alter cytokine release and inflammatory mediator production, potentially affecting the immune system’s capacity to recognize and eliminate tumor cells. Given P2X7’s crucial role in immune regulation and its involvement in tumor cell apoptosis, targeting this receptor presents a promising strategy for cancer therapy aimed at restoring immune activity within the tumor microenvironment. Additionally, P2X7 receptor is well-known for its ability to form membrane pores and mediate calcium influx upon ATP binding. The interaction of Ap4A with this receptor activates several pathways, such as the AMP-activated protein kinase (AMPK) signaling cascade. AMPK, a key regulator of energy homeostasis, becomes activated in a calcium- and ROS-dependent manner via P2X7 in microglia and macrophages. This mechanism underlies metabolic reprogramming that supports cellular adaptation under stress, highlighting Ap4A’s role as a metabolic “alarmone.” For instance, tumor cells—often residing in a hypoxic or nutrient-depleted microenvironment—leverage this adaptability through P2X7-mediated upregulation of glycolytic enzymes and more efficient utilization of intracellular glycogen stores [45]. For instance, inhibiting P2X1R and P2X7R has been observed to reduce mitochondrial activity, intracellular calcium levels, and proliferation in various cancer cell lines, such as leukemia cell lines [46].

#### 4.5.5. P2X3 and P2X5 Receptors

In addition to the more widely studied P2X1, P2X4, and P2X7 receptors, P2X3 and P2X5 receptors have also been shown to play roles in pain signaling and tissue remodelling, which can be relevant in cancer biology. Although less is known about the direct interaction between Ap4A and these receptor subtypes, their activation in the context of cancer pain and tissue invasion could influence tumor progression and metastasis. P2X5, in particular, is associated with cell proliferation and may contribute to the unchecked growth seen in certain cancer types [47].

#### 4.5.6. P2Y Receptors

P2 receptors are widely expressed across various cancer types, with their activation or inhibition influencing tumor cell survival and proliferation. These findings have sparked growing interest in targeting P2 receptor signaling as a potential therapeutic strategy for cancer treatment [48]. P2Y receptors are a family of G protein-coupled receptors (GPCRs) that respond to extracellular nucleotides such as ATP, ADP, UTP, and UDP. Unlike ionotropic P2X receptors, P2Y receptors mediate their effects through second messenger systems, triggering a variety of intracellular responses [49]. There are eight known subtypes of P2Y receptors: P2Y1, P2Y2, P2Y4, P2Y6, P2Y11, P2Y12, P2Y13, and P2Y14, which are involved in physiological processes such as platelet aggregation, immune response, and regulation of vascular tone. Extensive research highlights the strong association between platelets and cancer progression.

Ap4A’s interaction with P2Y receptors significantly impacts cancer cell behaviour by modulating signaling pathways that control proliferation, migration, and apoptosis. For example, Ap4A acts as an agonist at P2Y1 receptors, which are involved in regulating cancer cell proliferation and migration [50]. This interaction is critical in modulating intracellular pathways that promote tumor progression. Similarly, P2Y2 receptors, which respond to ATP and UTP, regulate cell survival and apoptosis. When Ap4A binds to P2Y2, it influences these processes, potentially affecting tumor growth and resistance to cell death [51]. Moreover, the dual role of Ap4A is particularly evident in ocular physiology. Castany et al. [52] identified Ap4A and Ap5A as soluble components of the human aqueous humour and reported significantly elevated Ap4A levels in patients with glaucoma compared to healthy individuals. In this context, Ap4A was found to increase intraocular pressure (IOP) by stimulating P2Y2 receptors on the ciliary body, a mechanism associated with disease progression. Paradoxically, other studies have demonstrated that Ap4A can also lower IOP by activating P2Y6 receptors in the ciliary processes, suggesting a protective role in glaucoma models [53]. These contrasting effects underscore the complexity of Ap4A’s biological functions and highlight the importance of receptor-specific signaling in determining its physiological or pathological outcomes.

P2Y12R, plays a key role in platelet activation, facilitating aggregation and adhesion to cancer cells. This process supports extravasation and metastasis by enabling cancer cells to navigate the vascular system more effectively. Currently, P2Y12R inhibitors are primarily used as antiplatelet agents. However, their impact on tumor progression is complex and dual-faceted. While some studies suggest that inhibiting P2Y12R suppresses tumor growth and metastasis, others indicate that P2Y12R inhibitors may, in certain contexts, facilitate tumor development. Given these contrasting effects, careful evaluation is necessary when considering P2Y12R inhibitors for anticancer applications, balancing their potential benefits with associated risks. In the context, the interaction between P2Y12 receptors and Ap4A may influence metastasis by affecting platelet–tumor cell interactions, which are crucial for the dissemination of cancer cells to distant sites [54]. Moreover, diadenosine tetraphosphate holds therapeutic promise for patients at high risk of coronary thrombosis due to their rapid, reversible action and intravenous administration [55].

Ap4A can also modulate the tumor immune microenvironment by activating P2Y receptors, influencing immune cell infiltration and activity. By altering these immune interactions, Ap4A may affect the anti-tumor immune response and contribute to immune evasion by the tumor [56]. In addition to its role in cancer, Ap4A has been shown to interact with P2Y1 receptors in smooth muscle regulation. It has been demonstrated that Ap4A activates P2Y1 receptors in the circular smooth muscle of the mouse colon, causing muscle hyperpolarization and inhibition of spontaneous contractions [57].

In summary, Ap4A interacts with both P2X and P2Y receptors, influencing various aspects of cell biology (Table 1). These interactions affect processes such as angiogenesis, tumor growth, immune responses, and metastatic potential, underscoring the complex role of Ap4A in the TME. Targeting Ap4A and its associated signaling pathways offers promising avenues for developing anticancer therapies aimed at disrupting tumor-supporting interactions and enhancing the efficacy of existing treatments.

### 4.6. Interaction with Other Proteins

Thanks to advancements in technology, unusual and novel Ap4A roles in metabolic events have been explored. Recent studies have shown that this alarmone may play multiple roles in cellular processes by interacting with and modifying other proteins. These findings, which could have potential therapeutic applications, will be briefly discussed to highlight emerging directions in the field.

A significant upregulation of Ap4A concentration was observed in dendritic cells (DCs) in an in vivo model using CD11c^+^ mice, which lack Ap4A hydrolase. DCs are immune cells known for their advanced migratory abilities, and they activate CD8^+^ T cells during an immune response by presenting antigens in secondary lymphoid organs [72]. In addition, it was previously reported that Ubiquitin-like Modifier-Activating Enzyme 1 (UBA1) is capable of synthesizing Ap4A in vitro [10]. Chemical proteomic profiling also revealed a variety of proteins involved in gene expression, translation, and protein targeting that interact with diadenosine triphosphate and tetraphosphate. UBA1, an essential enzyme in the ubiquitin system, was among the most significantly enriched hits for Ap4A [73]. In addition to Ap4A’s previously discussed effects on P2X and P2Y receptors, an increase in lactoferrin levels has also been noted. A single application of Ap4A has been shown to increase lactoferrin production, a protein known for its significant anti-inflammatory and antimicrobial properties [74]. Therefore, diadenosine tetraphosphate appears to support immune homeostasis on the ocular surface and may strengthen the tear film barrier.

Furthermore, it was recently demonstrated that Ap4A is a superior co-substrate for the auto-AMPylation of the FIC enzyme (FICD) in vitro. While FICD has been implicated in neurogenesis, its best-characterized in vivo target remains BiP, linking AMPylation to the cellular stress response. Notably, the authors found that FICD utilizes Ap4A more efficiently than ATP for auto-AMPylation, leading to a complete conversion of the enzyme into its AMPylated form, whereas ATP resulted in only partial AMPylation, leaving residual non-AMPylated FICD. These findings connect AMPylation to Ap4A alarmone formation as a novel cellular stress response mechanism [75]. Another study showed that Ap4A could act as a diabetogenic factor, inducing a diabetic state by decreasing Glucose Transporter Type 4 (GLUT-4), a protein responsible for glucose transport into cells, and reducing glucose uptake. These findings are particularly interesting as diadenosine polyphosphates have been shown to affect blood pressure, and the link between hypertension and diabetes remains unclear [76]. Given that cancer cells require more glucose to fuel their rapid growth, downregulation of GLUT-4 could hinder glucose uptake, potentially limiting tumor cell proliferation.

## 5. AP4A-Role in Gene Expression Regulation

### 5.1. Transcriptional Control

Beyond its involvement in stress response and metabolic regulation, Ap4A has emerged as a key modulator of gene expression, influencing transcriptional activity through both direct and indirect mechanisms. Ap4A plays a crucial role in transcriptional regulation by influencing key protein interactions, particularly those involving transcription factors that govern cancer-related pathways. One of the main interactions influenced by Ap4A is with the Histidine Triad Nucleotide Binding Protein 1 (HINT1) and Microphthalmia-associated Transcription Factor (MITF). MITF is crucial for regulating genes involved in melanocyte development, pigmentation, and the progression of melanoma. Ap4A has been reported to disrupt the formation of the HINT1-MITF complex, a critical step in MITF’s transcriptional activation. This disruption leads to attenuation of MITF-mediated gene expression, thereby suppressing melanoma cell growth [77]. Simply increasing Ap4A releases HINT-1 from MITF and activates transcription of MITF-targeted genes, establishing Ap4A as an important second messenger for transcriptional activation [78].

Moreover, the impact of Ap4A on gene expression has been validated through transcriptomic studies. In chronic myelogenous leukemia cells lacking the NUDT2 hydrolase—an enzyme responsible for degrading Ap4A—a dramatic increase in intracellular Ap4A (up to 175-fold) led to transcriptional changes in over 6000 genes. These changes notably included the downregulation of genes involved in tumorigenesis and immune responses, reinforcing Ap4A’s ability to function as a global regulator of gene expression.

Recent studies have shown that the Fragile Histidine Triad Diadenosine Triphosphatase (FHIT), a tumor suppressor gene, binds Ap4A, although with lower affinity compared to Ap3A, hydrolysing it into ATP and AMP [79]. The FHIT protein plays a critical role in regulating cell proliferation and apoptosis, and its loss is among the most frequent copy number alterations in cancers, particularly in digestive tract tumors [80]. These findings suggest that FHIT-mediated hydrolysis of Ap4A may represent a crucial mechanism through which this tumor suppressor modulates cellular stress responses and maintains genomic stability, with its inactivation potentially contributing to tumorigenesis via dysregulated Ap4A signaling.

Together, these findings highlight the context-specific regulatory roles of Ap4A in cancer biology (Figure 3).

### 5.2. AP4A-Mediated Post-Transcriptional Gene Expression Regulation

RNA capping is an essential modification process that occurs during the synthesis of messenger RNA (mRNA) in eukaryotic cells. This modification involves the addition of a cap structure at the 5’ end of the nascent RNA molecule, which is critical for RNA stability, efficient translation, and proper export from the nucleus to the cytoplasm. The cap structure, typically a 7-methylguanylate cap (m7G cap), is a modified guanine nucleotide that is added in a series of enzymatic steps [81]. Thus, RNA capping has a protective and functional importance as the 5’ cap protects the mRNA from degradation by exonucleases. Without this cap, mRNA would be rapidly degraded in the cell, preventing the translation of essential proteins [82].

Depending on the mode of hydrolysis and whether it functions only on the non-canonical cap (NCC) RNA or also on the nucleotide metabolite, the NCC decapping enzymes can be divided into four classes. Thus, the Ap4A hydrolase is one of the decapping class enzymes, which releases ppRNAs or pRNAs depending on the length of the polyphosphate linker. It has decapping activity toward dinucleotide polyphosphate caps and can also hydrolyze the dinucleotide polyphosphate itself [83].

Diadenosine tetraphosphate introduces an alternative capping mechanism that deviates from this conventional pathway, adding another layer of regulation in gene expression, particularly during stress responses. One of the intriguing roles of elevated Ap4A during stress, in fact, is its ability to be incorporated into the 5’ end of RNA, replacing the conventional m7G cap [84]. This incorporation results in the formation of an alternative capping mechanism which is thought to be a protective measure that helps the cell manage its resources more effectively during stress. The Ap4A cap alters the fate of the capped mRNA, primarily by affecting its translation efficiency. Unlike the m7G cap, which facilitates efficient translation initiation by being recognized by the cap-binding protein eukaryotic initiation factor 4E (eIF4E), the Ap4A cap leads to selective translation [85]. mRNAs with an Ap4A cap are often translated less efficiently, which may seem counterintuitive at first [86]. However, this reduced translation is beneficial under stress conditions as it helps conserve cellular energy and resources. By limiting the translation of certain mRNAs, the cell can prioritize the synthesis of proteins that are critical for stress survival, such as heat shock proteins and other stress-responsive factors [87]. This selective translation mediated by the Ap4A cap ensures that the cell can quickly adapt to changing conditions, maintaining homeostasis and enhancing survival prospects. Furthermore, the alternative capping may also influence mRNA stability and decay rates, thereby fine-tuning the expression of stress-responsive genes at multiple levels. The presence of this alternative cap could also potentially alter the interactions between mRNA and RNA-binding proteins, further modulating the post-transcriptional regulation of gene expression during stress [88]. In conclusion, Ap4A serves as a crucial player in the alternative capping of RNA, especially under stress conditions.

By promoting the formation of Ap4A-capped mRNAs, Ap4A helps orchestrate a selective translation response that is pivotal for cellular adaptation and survival in the face of environmental challenges. The alternative capping mechanism introduced by Ap4A represents a significant regulatory pathway in gene expression, particularly under stress. By incorporating Ap4A at the 5’ end of RNA, cells adapt to stress by altering mRNA fate and translation efficiency. Unlike the conventional m7G cap, which promotes robust translation, the Ap4A cap supports selective and reduced translation, conserving cellular resources during stress. In cancer, this mechanism plays a dual role. Elevated Ap4A levels in tumors could protect cancer cells during metabolic or environmental stress, supporting survival and adaptation. However, the selective translation promoted by the Ap4A cap might also limit the synthesis of certain proteins essential for tumor suppression. These dynamics make the Ap4A capping pathway a potential therapeutic target, as modulating this process could disrupt the stress adaptations critical for cancer progression.

For instance, it has been demonstrated that under disulfide stress, there is a marked increase in Np4A levels (N = adenosine, guanosine, uridine, or cytidine), which is accompanied by widespread Np4 capping of E. coli transcripts. This results in prolonged RNA lifetimes, suggesting a mechanism through which dinucleotide tetraphosphates may act as second messengers in cellular stress responses, a role potentially conserved in eukaryotes [89].

## 6. The Therapeutic Potential of Ap4A

As outlined in the updated framework of cancer hallmarks, the functional pleiotropy of Ap4A extends to critical processes such as immune evasion and stress tolerance, reinforcing its potential as a multifaceted therapeutic target [90]. Diadenosine tetraphosphate has emerged as a multifaceted signaling molecule whose biological functions span from cell proliferation to immune modulation and metabolic adaptation. Its widespread influence appears to be largely mediated through interactions with purinergic receptors—particularly the ionotropic P2X7 receptor, which plays a central role in orchestrating many downstream effects observed across different tissues and pathological contexts.

As an alarmone, Ap4A signals cellular distress and can initiate immune responses, enhancing innate immunity in the context of infections and malignancies. Importantly, the mechanisms governing Ap4A synthesis and degradation are not uniform across tissues; instead, they are highly context-dependent, shaped by each tissue’s physiological demands and metabolic status. In cancer and immune cells, Ap4A regulation is particularly distinct, involving upregulated synthesis or altered degradation pathways—an aspect that offers unique opportunities for targeted therapeutic intervention, especially given the frequent dysregulation of Ap4A homeostasis in tumors.

As previously discussed, the link between Ap4A, P2X7, and cancer progression is compelling. Its capacity to modulate extracellular ATP signaling—possibly by influencing ATP release or degradation—may indirectly affect P2X7 receptor activity and downstream effects within the tumor microenvironment (TME) [44]. Moreover, P2X7 not only confers survival advantages to cancer cells but also contributes to tumor angiogenesis. Among P2X receptors, P2X7 uniquely promotes the secretion of vascular endothelial growth factor (VEGF)—a critical driver of neovascularization. VEGF release has been confirmed in several cancer cell lines and other cell types such as monocytes and HEK293 cells expressing P2X7 [91]. Given its dual function in metabolic adaptation and angiogenesis, P2X7 represents a promising therapeutic target in oncology. Thus, inhibitors of P2X7 or strategies aimed at disrupting Ap4A-mediated signaling may hold potential for controlling tumor progression, immune suppression, and therapeutic resistance by modulating purinergic signaling dynamics within the TME.

Moreover, Ap4A plays a role in angiogenesis by acting as an agonist at P2X1 and P2X4 receptors, stimulating endothelial cell proliferation and migration—key events in neovascularization [43,92]. Beyond immune modulation, Ap4A analogs may also enhance immunotherapies or vaccine efficacy, particularly in immunocompromised patients. As tumors rely on neovascularization for growth and nutrient supply, targeting Ap4A-related pathways may offer novel anti-angiogenic strategies. Indeed, clinical investigations have already established P2X receptor antagonists as potential treatments for chronic cough and cancer-related pain, providing a foundation for further exploration of their application in managing other cancer-associated symptoms [42]. Recent studies have also synthesized Ap4A analogs with dual inhibitory activity at P2Y1 and P2Y12 receptors, demonstrating synergistic inhibition of platelet aggregation and underscoring Ap4A’s potential as an antithrombotic agent [93]. These inhibitory properties make also Ap4A analogs attractive candidates for drug development in cardiovascular medicine.

In ophthalmology, Ap4A has shown efficacy in reducing intraocular pressure (IOP) in glaucoma models, potentially through activation of P2Y6 receptors in the ciliary body, highlighting another therapeutic dimension [53].

Interestingly, Ap4A levels exhibit marked fluctuations depending on the cellular environment. Under growth-inhibitory conditions, such as amino acid starvation or serum deprivation, Ap4A concentrations drop significantly—while ATP remains relatively stable. This dissociation underscores Ap4A’s role as a specific stress signal rather than a mere energy reserve analog. Targeting Ap4A or its signaling pathways may offer novel therapeutic avenues not only in oncology but also in the treatment of chronic inflammation and stress-related disorders. Its role in the TME, especially in modulating immune evasion, warrants further exploration, particularly in the context of immunotherapy, where it could enhance the efficacy of immune-based treatments. Ap4A’s involvement in stress responses and RNA capping mechanisms can be leveraged to develop treatments for diseases characterized by cellular stress and dysregulated protein synthesis. Ap4A’s incorporation into the RNA cap structure helps modulate the translation of stress-responsive genes [94]. This adaptive response is crucial for cell survival under adverse conditions. Therapeutic modulation of Ap4A levels or its interaction with the RNA capping machinery could help manage diseases with disrupted stress response pathways.

Furthermore, its regulatory potential has been demonstrated through transcriptomic analyses, particularly in chronic myelogenous leukemia (CML) models. In CML cells deficient in NUDT2, the enzyme responsible for hydrolyzing Ap4A, intracellular levels of this dinucleotide were shown to rise dramatically. This substantial accumulation triggered widespread transcriptional reprogramming, altering the expression of more than 6000 genes. Crucially, many of these changes included the downregulation of genes implicated in tumor progression, immune evasion, and metabolic reprogramming, suggesting that elevated Ap4A levels can act as a transcriptional “brake” on cancer-promoting pathways. This broad modulatory effect underscores Ap4A’s role not just as a signaling molecule, but as a global regulator of gene networks that are essential for malignant transformation and proliferation. From a therapeutic standpoint, this opens new opportunities: targeting Ap4A metabolism—either by inhibiting NUDT2 or administering stable Ap4A analogs—could modulate gene expression in a way that suppresses tumorigenesis or sensitizes cancer cells to treatment. Moreover, given Ap4A’s natural presence and diverse physiological roles (e.g., in immune modulation, angiogenesis, and stress response), therapeutics based on Ap4A may offer multi-target benefits with fewer systemic toxicities.

Combining Ap4A-targeting therapies with existing cancer treatments, such as chemotherapy, radiation, or immune checkpoint inhibitors, could provide synergistic effects, enhancing the overall efficacy of cancer treatment. For instance, targeting Ap4A-mediated metabolic pathways could sensitize cancer cells to chemotherapy by reducing their ability to adapt to the metabolic stress induced by treatment [95].

Furthermore, a model of Ap4A in promoting aminoglycoside killing was suggested. Treatment of bacteria with aminoglycosides leads to the production of misfolded proteins, which cause the disruption of the bacterial membrane and the generation of ROS. Therefore, ROS induces LysU expression and its production of Ap4A. Diadenosine tetraphosphate might, in turn, inhibit intracellular proteostasis and aggravate the disruption of the membrane thereby promoting bacterial cell death [38]. Certain bacterial infections are associated with cancer development due to their ability to cause chronic inflammation, induce DNA damage, and affect immune responses. Thus, this novel potential activity is useful.

Given its involvement in cellular stress signaling and proliferation, Ap4A emerges as both a potential biomarker and a therapeutic target in cancer. Modulating its levels—through regulation of synthesis or hydrolysis—could reprogram tumor metabolism, restore tumor suppressor activity, and reshape gene expression profiles. However, the precise molecular mechanisms by which Ap4A exerts these effects remain to be fully elucidated, particularly regarding whether its influence is universally pro-tumorigenic or context-dependent [44].

While preclinical studies underscore Ap4A’s therapeutic promise, especially in cancer biology, more comprehensive clinical research is essential to validate its application and optimize its integration into therapeutic regimens.

## 7. Conclusions

Ap4A represents both a challenge and an opportunity in cancer therapy. Its role in promoting tumor cell survival under stress suggests that targeting its pathways could enhance therapeutic efficacy, yet its involvement in fundamental processes like DNA repair and metabolic adaptation necessitates a careful approach to avoid unintended consequences. Modulating Ap4A’s interaction with purinergic receptors in the TME could offer new strategies to enhance immune surveillance or inhibit tumor progression. Similarly, its role in alternative RNA capping presents a potential avenue for disrupting stress-induced translation programs that drive treatment resistance. In ovarian and melanoma cancers, Ap4A functions largely as a suppressor of proliferation and transcriptional activation, whereas in gastrointestinal cancers, dysregulation of its metabolism through loss of FHIT may promote tumor progression. These mechanisms underscore Ap4A’s potential as both a biomarker and therapeutic target, particularly in strategies aiming to modulate its synthesis, degradation, or receptor-mediated signaling.

To fully harness Ap4A’s therapeutic potential, future research must elucidate the molecular mechanisms regulating its synthesis, degradation, and cellular interactions. This will enable the development of precise modulators with minimal off-target effects, allowing for tailored interventions based on cancer type and disease stage. Additionally, clarifying whether ApnA (n = number of phosphates) molecules act as functional regulators or merely accumulate as metabolic by-products is essential for determining their true significance in tumor biology.

In summary, Ap4A’s multifaceted roles make it a compelling target for future cancer therapies. Addressing the current controversies and unanswered questions will be critical to translating its biological complexity into actionable clinical strategies, potentially paving the way for innovative treatments that improve patient outcomes.

## Figures and Tables

**Figure 1 molecules-30-03056-f001:**
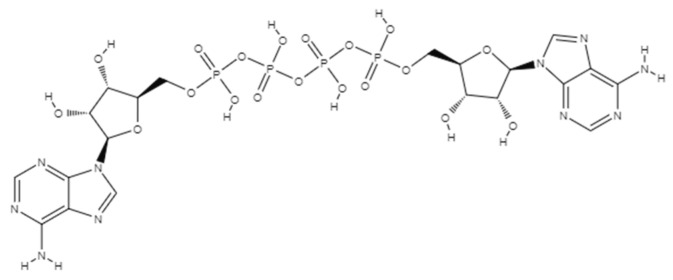
The structure of diadenosine tetraphosphate (Ap4A).

**Figure 2 molecules-30-03056-f002:**
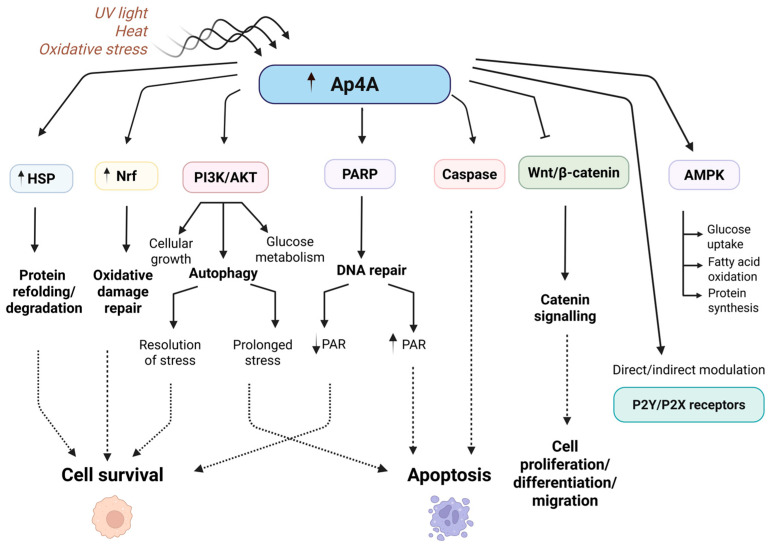
Ap4A as a central regulator of stress response, metabolism, and cell fate decisions. This scheme illustrates the multifaceted roles of diadenosine tetraphosphate (Ap4A) in cellular stress adaptation and signaling. Upon exposure to different stressors intracellular levels of Ap4A increase, triggering various downstream pathways. Altogether, this figure underscores Ap4A’s central role in orchestrating diverse signaling networks that determine cell fate between survival and apoptosis.

**Figure 3 molecules-30-03056-f003:**
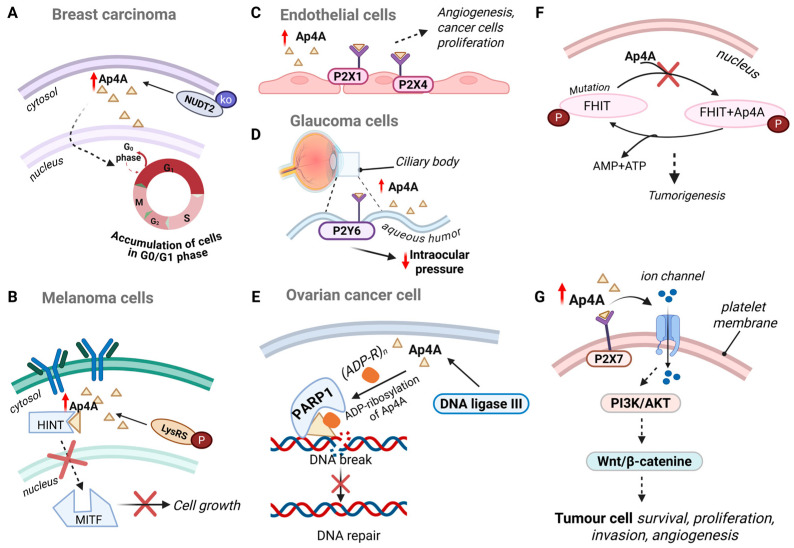
Simplified examples of a context-specific regulatory roles of Ap4A in cancer biology. (**A**) In breast carcinoma cell lines NUDT2 knock out can lead to accumulation of Ap4A in cells and blocking the cell cycle in the G_0_/G_1_ phase, stopping cell growth. (**B**) Ap4A can disrupt the formation of the HINT1-MITF complex, a critical step in MITF’s transcriptional activation, leading to attenuation of MITF-mediated gene expression, thereby suppressing melanoma cell growth. (**C**) Ap4A acts as an agonist at P2X1 receptors, which are predominantly expressed on endothelial cells. The activation of P2X1 by Ap4A can promote endothelial cell proliferation and migration, key processes in angiogenesis, which is crucial for tumor growth as it provides the blood supply necessary for rapid cancer cells proliferation. (**D**) Elevated Ap4A levels can lower IOP by activating P2Y6 receptors in the ciliary processes, suggesting a protective role in glaucoma models. (**E**) DNA ligase III synthesizes Ap4A, which acts as a primer for ADP-ribosylation by PARP. The ADP-ribosylated Ap4A binds to the protein associated with DNA polymerase-a-primase and inhibits it, reducing its interaction with DNA and thus, preventing DNA repair in ovarian cancer cells. (**F**) The FHIT protein, a tumor suppressor, binds and hydrolyzes Ap4A converting it into ATP and AMP. In many cancers, FHIT is mutated, resulting in disrupted Ap4A homeostasis, leading to tumorigenesis. (**G**) Ap4A may influence P2X7 receptor’s function either directly at high concentrations or indirectly through the modulation of extracellular ATP dynamics. Activation of P2X7 in tumor cells has been shown to trigger a range of downstream signaling pathways, including PI3K/AKT, JNK, and Wnt/β-catenin cascades, which collectively support tumor cell survival, proliferation, invasion, and angiogenesis.

**Table 1 molecules-30-03056-t001:** Summary of Ap4A interactions with purinergic receptors and their implications.

Receptor	Type	Ap4A Role	Cancer Implications	Therapeutic Avenues	Ref.
P2X1	Ionotropic	Acts as an agonist; promotes endothelial cell proliferation and migration	Facilitates angiogenesis, supporting tumor growth by supplying oxygen and nutrients.	Proliferation of highly malignant T24 bladder cancer cells depended on autocrine signaling through P2X receptors	[41,58]
P2X3	Ionotropic	Possible interaction with cancer pain pathways	May influence tumor-induced pain and tissue remodelling.	P2X3 purinergic receptor overexpression is associated with poor recurrence-free survival in hepatocellular carcinoma patients	[59,60]
P2X4	Ionotropic	Activates endothelial cells; enhances pro-angiogenic cytokine release	Promotes vascularization within tumors, aiding their survival in hypoxic environments.	P2X4R activation enhances the invasiveness of prostate cancer	[58,61]
P2X5	Ionotropic	Supports tissue proliferation	Potentially contributes to uncontrolled tumor cell growth	P2X5 is highly expressed in a broad range of lymphoid malignancies	[47,62]
P2X7	Ionotropic	Indirect modulation; triggers pro-inflammatory cytokine release	Contributes to the tumor’s inflammatory microenvironment and immune evasion.	P2X7R was highly expressed in gastric cancer tissues and gastric cancer cells	[45,63]
P2Y1	Metabotropic	Agonist; regulates cell proliferation and migration	Modulates intracellular signaling pathways critical for tumor progression.	Novel P2Y1 receptor ligands as a potent anti-prostate cancer agent	[64,65]
P2Y2	Metabotropic	Regulates cell survival and apoptosis	Promotes tumor resistance to cell death and enhances survival under stress conditions.	P2Y2 antagonism promoted apoptosis in HCC cells Human and murine oral cancer cell lines express numerous P2 receptors	[66,67]
P2Y6	Metabotropic	Activates during stress; influences metabolism and immune response	Modulates tumor adaptation and immune system interactions.	P2Y_6_R may represent a prime target for reducing colorectal carcinogenesis	[68,69]
P2Y12	Metabotropic	Affects platelet aggregation and tumor cell interactions	Facilitates metastasis by enabling dissemination via the circulatory system.	P2RY12 inhibitors could reduce the tumor spread in melanoma, ovarian, breast, lung, and pancreatic cancers	[70,71]

## Data Availability

The data presented in this review are based on previously published studies and are cited within the manuscript. No new data were created.

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
