# Peer review of "Ap4A in Cancer: A Multifaceted Regulator and Emerging Therapeutic Target"

_molecules, 2025, doi:10.3390/molecules30153056_

Round 1
Reviewer 1 Report
Comments and Suggestions for Authors
The manuscript demonstrates a clear overall structure and relatively thorough literature review. It is recommended for acceptance after minor revisions:
-
The Introduction section is too brief and does not adequately cover the research background. For instance, the current status of tumor therapy could be expanded, with references to relevant literature such as Chem. Commun., 2025, 61, 5333-5336; Chem. Commun., 2025, 61, 2484-2495; and Chin. Chem. Lett., 2023, 34, 108088.
-
The section on AP4A structural conformation (currently a separate subsection) could be integrated into the Introduction to reduce the number of subheadings.
-
Page 3, Line 96: The correct subheading should be "3.2 Degradation of Ap4A" (currently appears to be 1.2 Degradation of Ap4A").
-
The text in Figure 2 is too small and illegible. Please enlarge the font size or improve the resolution for clarity.
NO
Author Response
|
Summary |
|
|
|
Thank you very much for taking the time to review our review. Please find the detailed responses below and the corresponding revisions/corrections. Minor revisions as was suggested have been made.
|
||
|
Comments 1: The Introduction section is too brief and does not adequately cover the research background. For instance, the current status of tumor therapy could be expanded, with references to relevant literature such as Chem. Commun., 2025, 61, 5333-5336; Chem. Commun., 2025, 61, 2484-2495; and Chin. Chem. Lett., 2023, 34, 108088.
|
||
|
Response 1: We thank the reviewer for the insightful suggestion to expand the introduction section and for pointing out highly relevant and interesting literature. For instance, the Introduction was amplified, and all the modifications can be tracked in red ‘’ Ap4A functions as both a signalling molecule and a regulator of cellular homeostasis, …. in leveraging Ap4A’s biology for cancer treatment.’’ (page 1-2, line 35-71). However, we found the proposed references to be quite specific, especially considering that our manuscript does not address or mention antitumor compounds or nanomaterials in any section. For this reason, we have opted not to expand the literature in that specific direction to maintain the focus and coherence of the manuscript.
|
||
|
Comments 2: The section on AP4A structural conformation (currently a separate subsection) could be integrated into the Introduction to reduce the number of subheadings. |
||
|
Response 2: This is a good consideration. Nevertheless, since the introduction section was amplified, now the text and the separate heading is integrated more harmonically, so we preferred to maintain it as a separate subheading in order to preserve clarity and emphasize its specific relevance within the manuscript.
|
||
|
Comments 3: Page 3, Line 96: The correct subheading should be "3.2 Degradation of Ap4A" (currently appears to be 1.2 Degradation of Ap4A"). |
||
|
Response 3: Thank you to pointing this out, the numeration of subheading has been corrected.
|
||
|
Comments 4: The text in Figure 2 is too small and illegible. Please enlarge the font size or improve the resolution for clarity. |
||
|
Response 4: We are completely agreed, thus, the text inside the figure was enlarged.
|
||

Reviewer 2 Report
Comments and Suggestions for Authors
This is a well-conceived and timely review that attempts to summarize the emerging role of diadenosine tetraphosphate (Ap4A) in cancer biology. The authors have done a commendable job compiling a diverse range of Ap4A-associated cellular functions and signaling networks and linking them to cancer-relevant phenotypes. However, the manuscript would benefit from significant refinement to improve flow, eliminate redundancy, and elevate scientific depth in key areas. With revision, this article has the potential to serve as a useful resource for researchers exploring nucleotide signaling and its therapeutic implications in oncology.
Major Comments
- Conceptual Framework Needs Strengthening
The review would benefit from a clearer conceptual structure early on. Consider outlining the physiological role of Ap4A in normal cells versus its dysregulation in cancer, and then building on that foundation to address mechanistic and therapeutic relevance. - Expand Mechanistic Insights
The discussion of Ap4A's role in apoptosis, DNA repair, and stress responses remains somewhat descriptive. Incorporate specific mechanistic details and relevant pathways—for instance, how Ap4A levels modulate PARP1 activity, or its role in nucleotide excision repair and replication stress. Including more mechanistic insight will significantly strengthen the review’s value. - Cancer-Specific Relevance Needs Focus
While the authors mention several cancer hallmarks, the connection between Ap4A signaling and cancer progression is not always clearly articulated. It would be helpful to include cancer-specific examples or data where Ap4A levels have been directly measured in tumor tissues or linked to prognosis, therapy resistance, or immune evasion. - Therapeutic Targeting: Overlooked Opportunity
The therapeutic potential of modulating Ap4A metabolism or signaling is only briefly discussed. Please expand this section: - Mention inhibitors/activators of Ap4A-related enzymes (e.g., NUDT2, Ap4A hydrolases).
- Discuss feasibility of Ap4A analogs or small molecules modulating its pathways.
- Highlight whether Ap4A might serve as a biomarker in cancer diagnostics or treatment stratification.
- Figures and Tables
A simplified schematic summarizing Ap4A metabolism and its intersecting cancer-related pathways would greatly help readers. Consider a visual map to illustrate the enzymes, targets, and downstream effects of Ap4A in the cancer context.
Minor Comments
- Language and Redundancy: Several sections repeat information or use verbose language. A careful edit to streamline and remove redundancy will improve readability.
- Abbreviations: Define all abbreviations at first use and ensure consistency throughout the manuscript.
- Title Suggestion: Consider simplifying the title to improve impact, e.g., "Ap4A in Cancer: A Multifaceted Regulator and Emerging Therapeutic Target."
Author Response
|
Summary |
|
|
|
Thank you very much for taking the time to review this manuscript. Your comments are really insightful and useful in order to clarify and harmonize the manuscript. Please find the detailed responses below and the corresponding revisions and corrections highlighted in red in the re-submitted files. We hope these revisions meet your expectations and contribute to a clearer and more robust scientific discussion. |
||
|
|
|
|
|
Comments 1: Conceptual Framework Needs Strengthening The review would benefit from a clearer conceptual structure early on. Consider outlining the physiological role of Ap4A in normal cells versus its dysregulation in cancer, and then building on that foundation to address mechanistic and therapeutic relevance.
|
||
|
Response 1: We thank the reviewer for their valuable feedback. In response to the suggestion to enhance the manuscript's depth and clarity, we have substantially amplified and extended the contextual content throughout the text. These additions provide further background, detail, and scientific rationale to support the discussed mechanisms and observations. To ensure full transparency and facilitate the review process, all newly added or expanded paragraphs have been clearly highlighted in red within the revised manuscript. We hope these improvements address the reviewer's concerns and contribute to a more comprehensive and informative version of the manuscript.
|
||
|
Comments 2: Expand Mechanistic Insights The discussion of Ap4A's role in apoptosis, DNA repair, and stress responses remains somewhat descriptive. Incorporate specific mechanistic details and relevant pathways—for instance, how Ap4A levels modulate PARP1 activity, or its role in nucleotide excision repair and replication stress. Including more mechanistic insight will significantly strengthen the review’s value.
|
||
|
Response 2: We thank you for your constructive comment. As per your suggestion, we have expanded and deepened several sections of the manuscript, particularly those related to the mechanisms involving Ap4A. These additions can be traced throughout the text and are highlighted in red for easier identification. For example, in section 4.4, page 7-8, line 303-314 “Cell cycle progression is a tightly regulated process… ‘’; and line 331-335 ‘’DNA ligase III synthesises Ap4A…’’. Additionally, different amplifications and insights can be found in section 4.5.4, page 9, line 387-393 ‘’Activation of P2X7 in tumour cells has been shown to trigger…’’ and line 400-409 of the same section ‘’P2X7 receptor is well-known for its ability to form membrane pores…’’. This addition aims to better clarify the central role of P2X7 in Ap4A-mediated processes, particularly in signal transduction and the regulation of cellular metabolism.
|
||
|
Comments 3: Cancer-Specific Relevance Needs Focus While the authors mention several cancer hallmarks, the connection between Ap4A signaling and cancer progression is not always clearly articulated. It would be helpful to include cancer-specific examples or data where Ap4A levels have been directly measured in tumor tissues or linked to prognosis, therapy resistance, or immune evasion.
|
||
|
Response 3: As suggested, different cancer-specific examples were cited in the text. For example, section 4.4, page 8, line 315 ‘’Thus, in breast carcinoma cell lines demonstrated that knocking down NUDT2 led to reduced cell proliferation and an accumulation of cells in the G0/G1 phase of the cell cycle, further highlighting Ap4A's role in regulating cancer cell growth.’’ Section 5.1, page 13, line ‘’ Ap4A has been reported to disrupt the formation of the HINT1-MITF complex, a critical step in MITF's transcriptional activation. This disruption leads to attenuation of MITF-mediated gene expression, thereby suppressing melanoma cell growth [77].’’ Section 5.1, page 12 line 531 ‘’In chronic myelogenous leukemia cells lacking the NUDT2 hydrolase—an enzyme re-sponsible for degrading Ap4A—a dramatic increase in intracellular Ap4A (up to 175-fold) led to transcriptional changes in over 6,000 genes.
Comments 4: Therapeutic Targeting: Overlooked Opportunity The therapeutic potential of modulating Ap4A metabolism or signaling is only briefly discussed. Please expand this section: - Mention inhibitors/activators of Ap4A-related enzymes (e.g., NUDT2, Ap4A hydrolases); - Discuss feasibility of Ap4A analogs or small molecules modulating its pathways; - Highlight whether Ap4A might serve as a biomarker in cancer diagnostics or treatment stratification.
Response 4: As requested, we have substantially expanded the section discussing Ap4A as a therapeutic target. In particular, we elaborated on its dual role in angiogenesis and tumor proliferation, highlighting the context-dependent effects of Ap4A via purinergic receptor signaling—especially its interaction with P2X7 and P2X1 receptors. The updated text now discusses how Ap4A influences endothelial cell migration and proliferation, its ability to modulate the tumor microenvironment, and the implications for targeted therapies—including the use of Ap4A analogs or modulators of its metabolic enzymes (e.g., NUDT2, FHIT). All the newly added content can be found in the section 6 and is highlighted in red to facilitate identification and review. - Section 6, page 16, line 652 ‘’ Thus, inhibitors of P2X7 or strategies aimed at disrupting Apâ‚„A-mediated signaling may hold potential for controlling tumor progression, immune suppression, and therapeutic resistance by modulating purinergic signaling dynamics within the TME’’. - Section 6, page 16, line 664 ‘’ Recent studies have also synthesized Apâ‚„A analogues with dual inhibitory activity at P2Y1 and P2Y12 receptors, demonstrating synergistic inhibition of platelet aggregation and underscoring Apâ‚„A’s potential as an antithrombotic agent.’’ - In this section, Ap4A’s potential as both a biomarker and therapeutic target, particularly in strategies aiming to modulate its synthesis, degradation, or receptor-mediated signaling was discussed.
|
||
Comment 5: Figures and Tables
A simplified schematic summarizing Ap4A metabolism and its intersecting cancer-related pathways would greatly help readers. Consider a visual map to illustrate the enzymes, targets, and downstream effects of Ap4A in the cancer context.
Response 5: A new figure-Figure 3 was create to evidence enzymes, targets, and downstream effects of Ap4A in the cancer context.
Comment 6:
- Language and Redundancy: Several sections repeat information or use verbose language. A careful edit to streamline and remove redundancy will improve readability.
- Abbreviations: Define all abbreviations at first use and ensure consistency throughout the manuscript.
- Title Suggestion: Consider simplifying the title to improve impact, e.g., "Ap4A in Cancer: A Multifaceted Regulator and Emerging Therapeutic Target."
Response 6:
- We fully agree with your comment regarding redundancy and verbose language. In response, several sections of the manuscript have been carefully revised and streamlined to enhance clarity and avoid repetition.
- Abbreviations have also been reviewed and harmonized throughout the manuscript. All abbreviations are now defined at first use, and consistency has been ensured. Some new entries may also be noted due to recent content additions.
- We appreciate your suggestion regarding the title. It has been simplified accordingly to improve its clarity and impact.

Reviewer 3 Report
Comments and Suggestions for Authors
Kateryna Tkachenko et al provided an in-depth analysis of Ap4A’s biosynthesis, degradation, protein targets, and molecular mechanisms. Overall, this manuscript is well-written and well-organized. However, I think this manuscript needs minor revision and is not suitable for publishing in Molecules in its current form. Here are some comments of this manuscript for authors.
1. The introduction section is too simple, the authors should rewrite the introduction section to 3-4 paragraphs.
2. Is there any difference between the synthesis and degradation of Ap4A in normal and tumor tissues? And the difference in Ap4A level in normal and tumor tissues should be stated.
3. The connection to cancer is underdeveloped. Is Ap4 Aassociated with genomic instability, immune evasion and chemotherapy resistance?
4. Was Ap4A exploited as a potential therapeutic target or biomarker in clinical practice?
5. The relationship between Ap4A and tumor development is intriguing but somewhat vague. The review should specify the role of Ap4A in different cancer types and processes (e.g., metastasis, drug resistance), or mechanistic pathways (e.g., stress response, metabolic reprogramming).
6. The phrase "Environmental stress" is too broad. Please specify key stressors (e.g., oxidative damage, nutrient deprivation) to clarify the contexts where Ap4A is most studied.
Author Response
|
Summary |
|
|
|
Thank you very much for taking the time to review this manuscript. Please find the detailed responses below and the corresponding revisions and changes evidenced in red inside the in the resubmitted text. |
||
|
|
||
|
Comments 1: The introduction section is too simple; the authors should rewrite the introduction section to 3-4 paragraphs.
|
||
|
Response 1: We agree with this comment. Therefore, the introduction was amplified and improved. All the changes are highlighted in red.
|
||
|
Comments 2: Is there any difference between the synthesis and degradation of Ap4A in normal and tumor tissues? And the difference in Ap4A level in normal and tumor tissues should be stated. |
||
|
Response 2: We have, accordingly, integrated some parts in the manuscript to answer to this question, thus in section 3.2, page 4, line 168, we added the statement about ap4A’s degradation:’’ In cancer cells, degradation is often reduced due to downregulation or mutation of hydro-lases, leading to accumulation of Ap4A, which can interfere with tumour suppressor (e.g., HINT1) and modulate gene expression.’’
|
||
Comment 3: The connection to cancer is underdeveloped. Is Ap4A associated with genomic instability, immune evasion and chemotherapy resistance?
Response 3: The connection to cancer was further emphasized and extensively discussed, highlighting the genomic involvement and introducing immune evasion mechanisms to explain the role of Apâ‚„A, for example in section 4.5.4, page9, line 393 ‘’Furthermore, P2X7 activation can lead to the formation of membrane pores, promoting cell death in immune cells and contributing to immune evasion by tumours. Thus, Ap4A’s indirect modulation of P2X7 receptors may alter cytokine release and inflammatory mediator production, potentially affecting the immune system’s capacity to recognize and eliminate tumour cells’’. Additionally, in section 4.4, page 8, line 329-335 ’’ Ap4A is synthesized under stress conditions, such as DNA damage, where it inhibits the initiation of DNA replication, contributing to the maintenance of genomic integrity [3,38]. DNA ligase III synthesises Ap4A, which acts as a primer for ADP-ribosylation by PARP. The ADP-ribosylated Ap4A binds to the Ap4A-binding site on the protein associated with DNA polymerase-a-primase and inhibits it, possibly by reducing its interaction with DNA. ADP-ribosylated Ap4A also diffuses to other replication forks and inhibits DNA polymerase-a-primase there’’.
Comment 4: Was Ap4A exploited as a potential therapeutic target or biomarker in clinical practice?
Response 4: Yes, the role of Ap4A as a potential target and/or biomarker has been now extensively explored and discussed in Section 6, page 15-17.
Comment 5: The relationship between Ap4A and tumor development is intriguing but somewhat vague. The review should specify the role of Ap4A in different cancer types and processes (e.g., metastasis, drug resistance), or mechanistic pathways (e.g., stress response, metabolic reprogramming).
Response 5: This very valuable suggestion. For instance, different parts discussing the relationship between Ap4A and tumor development have been integrated. Different parts now contain new paragraphs with mechanistic insight about the role of Ap4A. For example, in section 4.5.4, page 9, line 387-393 ‘’ Activation of P2X7 in tumor cells has been shown to trigger a range of downstream signalling pathways… facilitating tumour metastasis.’’ In addition, metabolic reprogramming is also widely discussed in section 4, page 4-5, line 174-185 ‘’ While traditionally considered a metabolic by-product,…with particular attention to the P2X7 receptor's role in cancer progression.’’ In section 4.2, page 5, line 214-216, new statements about stress response was added ‘’Ap4A accumulates under stress conditions such as oxidative stress, heat shock, and DNA damage… if it contributes to a stress response, elevates intracellular calcium, or causes mitochondrial dysfunction leading to ATP depletion.’’
Comment 6: The phrase "Environmental stress" is too broad. Please specify key stressors (e.g., oxidative damage, nutrient deprivation) to clarify the contexts where Ap4A is most studied.
Response 6: Specific examples of changes in Ap4a levels were added in section 3, page 4, line 136-146 ‘’ In Drosophila cell cultures, for instance, heat shock and heavy metal exposure led to a 2–3-fold increase in Apâ‚„A levels…the predominant mode of Apâ‚„A synthesis—whether constitutive or inducible, enzymatic or stress-regulated—varies according to cell type and functional context.’’

Reviewer 4 Report
Comments and Suggestions for Authors
In their paper, the authors reviewed the biological functions of diadenosine tetraphosphate (Ap4A), with particular emphasis on its role in cancerogenesis. Research in this area has been ongoing for decades, and the results have been periodically summarized in literature reviews. The review has sufficient novelty, as very few other review articles have focused on the role of Ap4A in human health and disease. I think that the authors have succeeded in creating a truly interesting review among many.
The manuscript is well-structured and clearly written. It provides a comprehensive overview of the latest research, with a thorough background. The sections provide comprehensive information. Additionally, the information has been properly organized in tables and illustrated.
I believe that the overall quality of the manuscript meets the high standards of the journal. However, I have several comments and suggestions. Please find them below.
- Lines 126-128 ‘A significant function of Ap4A lies in cell cycle regulation. By interacting with key cell cycle proteins, Ap4A influences cell division and growth, ensuring proper progression through the different phases of the cell cycle’.
I think it would be beneficial to shed some light on the mechanisms underlying cell cycle progression;
- The authors place a strong emphasis on the interaction between Ap4A and purinergic receptors. Although the roles of different receptors are reviewed in detail, I would suggest expanding the introduction to this section with a brief overview of the connection between purinergic receptors and cancer in general;
- In my opinion, it would be helpful to include information about which cells, organs, or tissues have shown a particular property of Ap4A. For instance, the authors mention the names of certain cell lines on line 333, but this information is not always provided for other cases.
In my opinion, the manuscript could be accepted with minor revisions.
Author Response
|
Summary |
|
|
|
Thank you very much for your thoughtful and encouraging comments on our manuscript. We truly appreciate you taking the time to review our work and are grateful for your recognition of the novelty and relevance of our focus on Ap4A, particularly in the context of cancerogenesis. It is very motivating to hear that you found our review both interesting and meaningful among the existing literature.
Please find below our detailed responses to your comments and suggestions, as well as the corresponding revisions highlighted in track changes in the re-submitted files. We hope the changes address your feedback effectively.
|
||
|
Point-by-point response to Comments and Suggestions for Authors
Comments 1: Lines 126-128 ‘A significant function of Ap4A lies in cell cycle regulation. By interacting with key cell cycle proteins, Ap4A influences cell division and growth, ensuring proper progression through the different phases of the cell cycle’. I think it would be beneficial to shed some light on the mechanisms underlying cell cycle progression.
|
||
|
Response 1: Thank you for pointing this out. We agree with this comment. Therefore, some insight about cell cycle progression was added in section 4.4, page 7, line 303-315 ‘’ Cell cycle progression is a tightly regulated process orchestrated by a sequential activation of CDKs and their regulatory partners… tumour cells with defective checkpoints continue to divide, accumulating mutations that drive onco-genesis’’
|
||
|
Comments 2: The authors place a strong emphasis on the interaction between Ap4A and purinergic receptors. Although the roles of different receptors are reviewed in detail, I would suggest expanding the introduction to this section with a brief overview of the connection between purinergic receptors and cancer in general. |
||
|
Response 2: Agree. Thus, the introduction section was widely revised and expanded in order to provide an overview about purinergic receptors and cancer in general. All new paragraphs are highlighted in red.
|
||
|
Comments 3: In my opinion, it would be helpful to include information about which cells, organs, or tissues have shown a particular property of Ap4A. For instance, the authors mention the names of certain cell lines on line 333, but this information is not always provided for other cases.
Response 3: Thank you for this helpful suggestion. We agree that specifying the cells in which particular properties of Ap4A have been observed adds valuable context and clarity for the reader. Thus, we have revised the manuscript to include additional examples of specific cancer cell lines associated with the described effects of Ap4A. These additions can now be found throughout the revised text, particularly in the relevant sections discussing Ap4A's role in cancer-related processes. The new information is highlighted in track changes for your convenience. |
||
|
- In Section 4.4, page 8, line 315-317: ‘’Thus, in breast carcinoma cell lines demonstrated that knocking down NUDT2 led to re-duced cell proliferation and an accumulation of cells in the G0/G1 phase of the cell cycle, further highlighting Ap4A's role in regulating cancer cell growth [36].’’ - In Section 4.4, page 8, line 325-327: ‘’This delay is believed to prevent replication, giving cells time to repair potential DNA damage, particularly in ovarian cancer cell lines [3].’’ - In Section 4.5.4, page 10, line 409-411:’’ For instance, inhibiting P2X1R and P2X7R has been observed to reduce mitochondrial ac-tivity, intracellular calcium levels, and proliferation in various cancer cell lines, such as leukemia cell lines [46].’’ - In Section 5.1, page 13, line 525-527: ‘’ Ap4A has been reported to disrupt the formation of the HINT1-MITF complex, a critical step in MITF's transcriptional activation. This disruption leads to attenuation of MITF-mediated gene expression, thereby suppressing melanoma cell growth [77].’’ |
||
|
|
||
|
|
||
